# Exploring the Effectiveness of Multi-Lingual Commonsense Knowledge-Aware Open-Domain Dialogue Response Generation

**Sixing Wu[1,2], Jiong Yu[1,2], Tianshi Che[3], Yang Zhou[3], and Wei Zhou[1,2]***

[1]National Pilot School of Software, Yunnan University, Kunming, China
[2] Engineering Research Center of Cyberspace, Yunnan University, Kunming, China
[3] Auburn University, Auburn, Alabama, USA
{wusixing, zwei}@ynu.edu.cn

## Abstract

Prior works have shown the promising results of commonsense knowledge-aware models in improving informativeness while reducing the hallucination issue. Nonetheless, prior works often can only use monolingual knowledge whose language is consistent with the dialogue context. Except for a few high-resource languages, such as English and Chinese, most languages suffer from insufficient knowledge issues, especially minority languages. To this end, this work proposes a new task, *Multi-Lingual Commonsense Knowledge-Aware Response Generation (MCKRG)*, which tries to use commonsense knowledge in other languages to enhance the current dialogue generation. Then, we construct a *MCKRG* dataset *MCK-Dialog*[1] of seven languages with multiple alignment methods. Finally, we verify the effectiveness of using multi-lingual commonsense knowledge with a proposed *MCK-T5* model. Extensive experimental results demonstrate the great potential of using multi-lingual commonsense knowledge in high-resource and low-resource languages. To the best of our knowledge, this work is the first to explore *Multi-Lingual Commonsense Knowledge-Aware Response Generation* .

## 1 Introduction

Open-domain dialogue response generation is always a tricky problem (Kann et al., 2022) because it allows users to converse in whatever they like, and then various knowledge will be involved (Zhou et al., 2022). Even with the expensive large-scale language models, models still suffer from the *non-informative* and *hallucination* problem if they can not access up-to-date and correct knowledge. The *non-informative* problem (Li et al., 2016) means the generated responses are always boring and generic, such as 'I don't know.'; the *hallucination* problem

---

*The corresponding author.

[1]Our codes and dataset are released at https://github.com/Y-NLP/Chatbots/tree/main/EMNLP2023_MCK-T5.

(Maynez et al., 2020) means factually incorrect knowledge that violates the real world may be generated; for example, 'iPhone 9 is my favorite.'.

Grounding dialogue response on the commonsense knowledge (Wu et al., 2020; Zhou et al., 2022; Cai et al., 2023) has shown great effectiveness in alleviating the aforementioned two problems. Large-scale commonsense knowledge such as ConceptNet (Speer et al., 2016) can provide various correct and informative real-world facts (Yu et al., 2020). Nonetheless, such works often can only leverage the knowledge of the native language. For example, an English system can only access the English knowledge base, and an Indonesian system can only access the Indonesian knowledge base. This restriction is not problematic for high-resource languages such as Chinese and English (Kann et al., 2022) because we can easily collect sufficient commonsense knowledge from the Internet. However, for many low-resource languages, collecting enough knowledge in their own language is exceptionally challenging. Without sufficient knowledge, knowledge-aware models face the problem of making bricks without straw, which would dramatically reduce their real-world practicalities (Wu et al., 2021, 2022b).

Given such challenges, this work proposes *Multi-Lingual Commonsense Knowledge-Aware Response Generation (MCKRG)*, a new task to break the bottleneck that models can only use the knowledge of the native language. The primary challenge of *MCKRG* is the dataset. As of the beginning of this work, only *KoWoW* is a large-scale English-Korean cross-lingual knowledge-aware dataset (Kim et al., 2021). However, *KoWoW* is not grounded on commonsense knowledge and only involves two languages. Thus, this work constructs a multi-lingual knowledge-aligned conversational dataset *MCK-Dialog* with several automatic processes. *MCK-Dialog* uses the *XPersona* of 7 languages as the conversational corpus; hence, *MCK-*

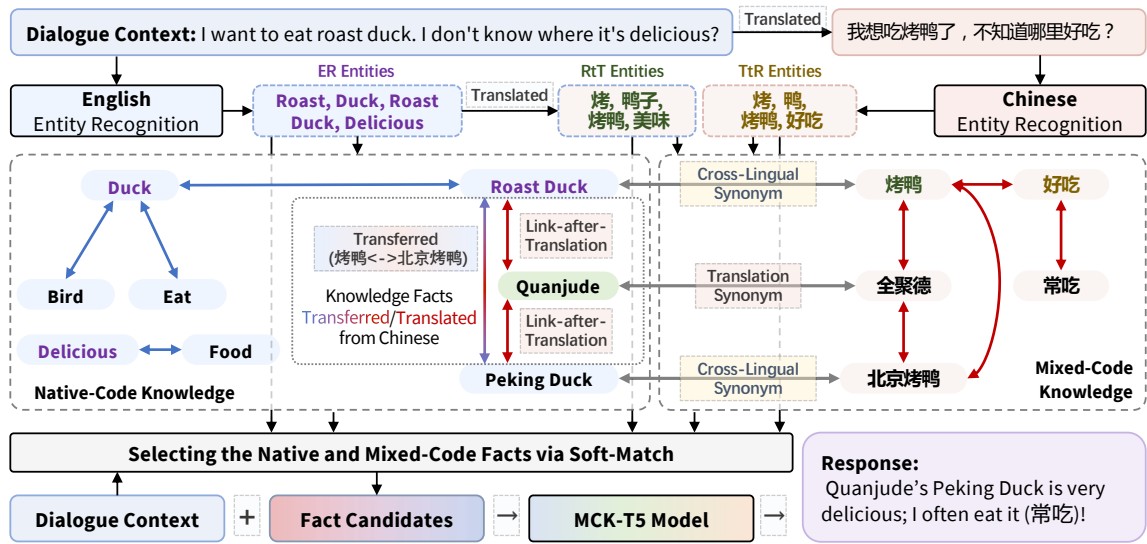

Figure 1: An illustration of *MCKRG*. In this example, *English* is the native language, and *Chinese* is used as an example to show how to use both the *native-code* and *mixed-code* multi-lingual knowledge.

*Dialog* simultaneously supports English (*En*), Chinese (*Cn*), French (*Fr*), Japanese (*Jp*), Korean (*Ko*), Italian (*It*), and Indonesian (*Id*).

As shown in Figure 1, we adopt *ConceptNet* (Speer et al., 2016) as the multi-lingual knowledge base. Besides the original *ConceptNet*, we have proposed two methods to transfer/translate multi-lingual knowledge into the current native language. Then, we propose one entity recognition method to retrieve entities in the native language and two methods to retrieve entities in other languages. Such entity recognition methods allow a model to find the knowledge of other languages easily. Last, we propose a *Hard Match* and a *Soft Match* to efficiently retrieve commonsense facts that can ground the current dialogue session.

Experiments use the proposed mT5-based (Xue et al., 2021) *MCK-T5* model to explore the effectiveness of *MCKRG*. Extensive results demonstrate using multi-lingual commonsense knowledge can significantly enrich the performance in low-resource and high-resource languages. Our further analyses also reveal how to use it correctly in the open-domain dialogue response generation. To sum up, the contribution of this work is three-fold:

1. To our best knowledge, we are the first to use multi-lingual commonsense knowledge to enhance dialogue generation (i.e., MCKRG);

2. We have constructed the first MCKRG benchmark dataset *MCK-Dialog* of 7 languages and propose a *MCK-T5* as the baseline;

3. Extensive experiments verified the effect of MCKRG and how to use multi-lingual commonsense knowledge in dialogue generation.

## 2 Related Work

**Knowledge-Aware Response Generation** Traditional response generation methods (Sutskever et al., 2014) often generates non-informative dialogues (Li et al., 2016) and factually incorrect hallucination information. To this end, knowledge-aware works ground dialogue response on external knowledge (Yu et al., 2020). The knowledge can be gathered from text-based encyclopedia (Lin et al., 2020a; Zhao et al., 2020), commonsense knowledge (Zhang et al., 2020; Wu et al., 2020; Zhou et al., 2022), search engines (Shuster et al., 2022), and many others (Moghe et al., 2020; Li et al., 2022). Although such works have achieved promising results, they often only focus on high-resource English or Chinese (Kann et al., 2022). Once in the low-resource language scenarios that knowledge is insufficient (Wu et al., 2022b), such works will degenerate into traditional non-knowledge-aware works. Compared to them, this work explores using multi-lingual commonsense knowledge besides the native language. In addition, although the prior work KoWoW (Kim et al., 2021) learns to use Korean-English cross-lingual knowledge, this work is totally different because: 1) we use graph-based commonsense knowledge instead of text-based knowledge; 2) KoWoW only studies a much simpler two language scenario.

| | En | Cn | Fr | Jp | Ko | It | Id |
|---|---|---|---|---|---|---|---|
| Train | 16.9K/124K *(The following units are also K)* | | | | | | |
| Valid | 1/7.3 | 0.22/1.7 | 0.25/1.9 | 0.28/2.1 | 0.3/2.3 | 0.14/1.1 | 0.48/3.8 |
| Test | 1/7.8 | 0.22/1.7 | 0.25/1.9 | 0.28/2.2 | 0.3/2.3 | 0.14/1.1 | 0.48/3.8 |

Table 1: The number of dialogue sessions/utterances.

| | *Native* | | | *Transferred* | | | *Translated* | | |
|---|---|---|---|---|---|---|---|---|---|
| | #E | #R | #F | #E | #R | #F | #E | #R | #F |
| En | 1.17M | 47 | 3.28M | 127K | 31 | 391K | 1.10M | 31 | 1.59M |
| Cn | 134K | 25 | 367K | 99.1K | 47 | 458K | 1.23M | 47 | 2.18M |
| Fr | 367K | 17 | 2.94M | 93.1K | 47 | 354K | 1.08M | 47 | 1.93M |
| Jp | 84.4K | 30 | 249K | 185K | 47 | 960K | 1.40M | 47 | 2.39M |
| Ko | 3.51K | 12 | 4.45K | 34.1K | 47 | 109K | 1.45M | 47 | 2.50M |
| It | 511K | 16 | 580K | 75.8K | 47 | 290K | 1.55M | 47 | 2.62M |
| Id | - | - | - | - | - | - | 1.53M | 47 | 2.62M |
| All | 3.95M | 47 | 9.51M | 590K | 47 | 2.56M | 7.56M | 47 | 15.8M |

Table 2: Knowledge abundance statistics of different sub-graphs. *#E/R/F* represents the number of distinct *entities*, *relations* and *facts* of each sub-graph.

**Multi-Source Knowledge** Prior works try to seek more knowledge from multiple sources to enrich the dialogue generation (Liang et al., 2021; Wu et al., 2021, 2022b). This work has a similar idea but the difference is also significant because we seek knowledge of other languages but the same knowledge type, while prior works seek knowledge of other knowledge types but the same language.

**Multi-Lingual** It allows a model to handle data of multiple languages in a uniform model. Using multi-lingual pre-trained language models such as mBART (Liu et al., 2020) and mT5 (Xue et al., 2021) are a convenient way. Although multi-lingual systems have been paid much attention in NLP, only a few works in dialogue response generation. MulZDG (Liu et al., 2022) and XDailyDialog (Liu et al., 2023) studies the non-knowledge-aware multilingual response generation, X-Persona (Lin et al., 2020b) studies the multi-lingual personalized response generation, and ToD (Majewska et al., 2023) studies the multi-lingual task-oriented response generation. Compared to such works, this work is the first work to study the multi-lingual commonsense knowledge-aware open-domain dialogue response generation.

**Cross-Lingual** For its primary issue, translation, there are three common paradigms (Huang et al., 2023; Zheng et al., 2023): 1) *translation-first* conducts translation before the generation while 2) *translation-later* conducts generation before the translation. The last 3) *end2end* does not explicitly conduct translation. This work has explored all paradigms: 1) Sec 3.2/Sec 3.3 uses the *translation-first/later* methods to transform knowledge of other languages to the current native language; 2) Our *MCK-T5* uses mixed-code knowledge in an *end2end* way.

## 3 MCK-Dialog Dataset

### 3.1 Conversations

*XPersona* (Lin et al., 2020b) is a multi-lingual extension of the monolingual English *Persona-Chat* (Zhang et al., 2018), which employed humans to generate dialogues grounded on an assigned persona. To obtain conversations in the other six languages, *XPersona* first used Neural Machine Translation (NMT) API with several human-in-the-loop mistake correction processes to construct the parallel training sets in other languages efficiently. Besides the NMT, the valid set and test set were well-revised by humans in the post-processing, which may change the dialogue or discard a dialogue; thus, as shown in Table 1, the valid/test sets are not in parallel across languages. *MCK-Dialog* keeps all dialogues of *XPersona*.

### 3.2 Commonsense Knowledge

We adopt the multilingual commonsense knowledge base *ConceptNet* (Speer et al., 2016). As reported in Table 2, we can see that the abundances of *Native* knowledge across languages are different. High-resource languages have 1M+ facts while low-resource languages can only have 4K+ facts, or even no facts. To enrich the abundance of knowledge, we leverage the knowledge of other languages. We create two expanded sub-graphs *Transferred* and *Translated* for each specific language $l_i$ besides the original *Native* sub-graph:

**Native:** For a fact $f = (e_{head}, e_{rel}, e_{tail})$ in the *ConceptNet*, if the language labels of $e_{head}$ and $e_{tail}$ are the same $l_i$, then it belongs to the *Native*$_{l_i}$.

**Transferred:** We transfer the knowledge of other languages to *Transferred*$_{l_i}$ by using the cross-lingual synonyms facts. For example, given two English-Chinese synonym facts, *(Apple Inc., Synonyms,* 苹果公司*)* and *(Tim Cook, Synonyms,* 蒂姆·库克*)*, then the English fact *(Apple Inc., Related To, Tim Cook)* can be transferred to a Chinese fact *(*苹果公司*, Related To,* 蒂姆·库克*)*.

**Translated:** Mainly for the low-resource and zero-resource languages. We employ the NMT API[2] to translate a fact from the source language to

---

[2]We use the Microsoft Azure Translation API.

the target language. Similar to the *Transferred*, we only translate the head entity and the tail entity. All newly translated facts are placed into the extension *Translated$_{l_i}$* for the specific language $l_i$.

Note that although both *Transferred$_{l_i}$* and *Translated$_{l_i}$* are built with multi-lingual knowledge, they still represent knowledge in the native language $l_i$. Hence, to avoid ambiguity, ***cross/multi-lingual*** is only used to tell the language relation between the dialogue context and the original knowledge source. Then, we use ***native-code*** to indicate the dialogue and knowledge are in the same language, and ***mixed-code*** to indicate the dialogue and knowledge may not be in the same language.

### 3.3 Knowledge Alignment

We then align the conversations with commonsense knowledge, which is a two-stage pipeline: the first *Entity Recognition* stage identifies entities, and the next *Fact Selection* stage selects fact candidates.

#### 3.3.1 Entity Recognition

We propose a native-code $ER_{l_{src}}$ to identify the entities of the native language and two mixed-code functions $TtR_{l_{src}\rightarrow tgt}$ and $RtT_{l_{src}\rightarrow tgt}$ to identify the entities of other languages.

**Native-Code** Given an utterance in language $l_{src}$, $ER_{l_{src}}$ first tokenizes and lemmatizes (optional) it to a n-gram list. Then, all stop-words are removed and we only retain the nouns, verbs, adjectives, and adverbs for all 1-gram candidate entities. Lastly, we filter the candidate entities using the entity set of the sub-graphs of the corresponding language $l_{src}$ to obtain the native-code entities. In the implementation, we employ several tools to handle each language. In detail, to tokenize a text, we use Natural Language Toolkit (NLTK) package (Bird et al., 2009) for English, French, Italian, and Indonesian; mecab-python3 package for Japanese; pymecab-ko package for Korean; splitting to characters for Chinese. Then, we use the same tool as the part-of-speech (POS) tagger for each non-Chinese language, and we directly remove all 1-gram Chinese entities. Furthermore, the Simplemma package is used as the lemmatizer for English, French, Italian, and Indonesian.

**Mixed-Code** The first $TtR_{l_{src}\rightarrow tgt}$ is translation-then-recognition style. It first translates the given utterance into the target language $l_{tgt}$ using the NMT API and then employs the corresponding $ER_{l_{tgt}}$ to identify the entities in the target language.

Differently, another recognition-then-translation style $RtT_{l_{src}\rightarrow tgt}$ first uses the native $ER_{l_{src}}$ to identify the entities, then uses the cross-lingual synonym facts (None-Indonesian) or NMT API (Indonesian) to obtain the entities in $l_{tgt}$.

#### 3.3.2 Fact Selection

Given a dialogue session $H = (X_1, \cdots, X_n)$ of $n$ turns and the identified entity set $(E_1, \cdots, E_n)$, we propose a *Hard Match* and a *Soft Match* to select the facts $K$ that ground the current session $H$.

**Hard Match** For a knowledge fact $f = (e_{head}, e_{rel}, e_{tail})$, if $e_{head} \in E_{1:t-1}$ while $e_{tail} \in E_t$, or $e_{tail} \in E_{1:t-1}$ while $e_{head} \in E_t$, then we assume $H$ is partially grounded on $f$.

**Soft Match** *Hard Match* method only considers the literal overlap but ignores the semantics, which may involve irrelevant knowledge. To this end, *Soft Match* first uses SentenceBERT (Reimers and Gurevych, 2020) to compute the fact embedding and the dialogue session embedding; then, *Soft Match* uses the Consine function to estimate the relevance in the embedding space. Then, facts are selected based on similarity scores.

### 3.4 Statistics

**Knowledge Abundance** As mentioned in Section 3.2, there is an obvious abundance gap between high-resource and low-resource languages. To bridge this gap, we transfer/translate facts from other languages to the current language. Table 2 reports the final knowledge abundance of three sub-graphs. It can be seen that *Transferred* can significantly expand the knowledge abundance for non-zero-resource languages. By using NMT APT, *Translated* can significantly enrich the knowledge abundance for not only the non-zero-resource languages but also zero-resource Indonesian. Nevertheless, we can not say *Translated* is better because it would introduce more noise in the translation.

**Knowledge Alignment** Figure 2 reports the average *matched* commonsense knowledge facts per dialogue session after the alignment. Besides, we also report the number of possible *related* facts, which only require that at least one entity appears in the dialogue session. We first focus on the native-code knowledge (i.e., *ER* bars). We can see that *En*, *Cn*, *Fr*, and *Jp* are high-resource languages because they have sufficient matched/related facts even if only use the *Naive* ConceptNet (i.e., *Native-ER*).

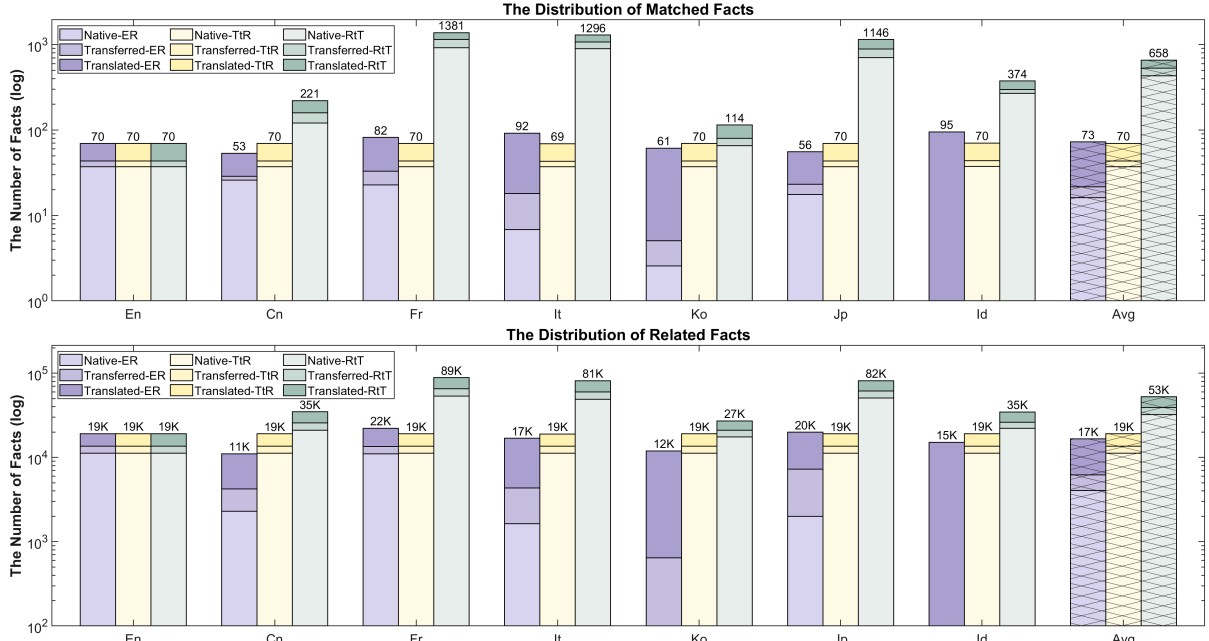

Figure 2: The distribution of *matched* and *related* commonsense facts per session. We separately report the counts of 1) the native-code facts using the *ER* to recognize entities; 2) the mixed-code facts using the *TtR* and *RtT*. In each bar, we separately report the distribution, where the lighted color is *Native*, the moderate is *Transferred*, and the darkest is *Translated*. Note that *Native-ER* shows the alignment status without using multi-lingual knowledge.

Unlike them, other languages are low-resource or zero-resource in the *Naive* ConceptNet. As for the next native-code knowledge *Transferred*, we can find its contribution (i.e., *Transferred-ER*) is limited for most languages after alignment. This is because its expansion relies on the cross-lingual synonym facts, which are also limited in ConceptNet. Unlike *Transferred*, the last native-code knowledge *Translated* can introduce more new knowledge facts, especially for the Indonesian (i.e., *Translated-ER*) with a higher tolerance for noise. By using all three sub-graphs, each language can have a similar knowledge abundance of the matched/related fact candidates. Then, we move to the mixed-code knowledge ( *TtR* and *RtT* bars). English have the same counts among the three bars because we only retrieve knowledge of English besides the native language. It can be seen that the translation-later style *RtT* can retrieve more knowledge facts than the translation-first style *TtR*.

Overall, it must be noted that all counts illustrated in Figure 2 are non-repetitive in each language and the total number of available facts should sum all sub-factors together. Consequently, the average number of matched/related facts per session is about 800 and 89K. Thus, similar to the construction of other knowledge-aligned datasets, although the retrieved facts are relevant to the context to

some extent, irrelevant knowledge still commonly exists, and this alignment is session-level. Therefore, selecting the most relevant knowledge at each turn of the dialogue generation is still needed.

## 4 MCK-T5 Model

To explore the effectiveness of *MCKRG*, we propose a *MCK-T5* Model, which is built upon the multi-lingual *mt5-base* (582M) (Xue et al., 2021). Formally, each instance is denoted as $(H_i, K_i, R_i)$, where $H_i = (X_{i,1}, \cdots, X_{i,n})$ consists of $n$-turns dialogue histories, $K_i = \{f_{i,j}\}^m$ consists of $m$ knowledge facts, and $R_i$ is the target response.

Since the backbone mT5 is a plain-text-based Encoder-Decoder network, we have to convert the structural knowledge facts $K_i$ into plain text and then concatenate the result with the dialogue history $H_i$. Thus, the input is formatted as follows:

$$Input = [Prompt_i; \Omega_K(K_i); \Omega_H(H_i)] \quad (1)$$

where $Prompt_i$ is an instruction code to the task, which is *'Generate a response based on the following terms:'*. The next $\Omega_K(K_i)$ linearizes all facts with identifiers; for example, *'Contextual Facts: [0] (Apple Inc., RelatedTo, Steven Jobs); ....; [10](iPhone, IsA, Device)'*. The last $\Omega_H(H_i)$ concatenates all dialogue histories with role labels,

just like *'Dialogue History: User: $X_1$; Bot: $X_2$; User: $X_3$...'*. mT5 does not need explicit labels to indicate the current language. Hence, we directly concatenate the $\Omega_K(K_i)$ and $\Omega_H(H_i)$ even if they are in different languages.

**Training** Thanks to the multi-lingual nature of mT5, this work considers two training strategies: 1) *Singleton*: We separately train a *MCK-T5* model for each language; 2) *Joint*: All languages share one same *MCK-T5* model in a specific scenario. For both strategies, we uniformly adopt the Maximum Likelihood Estimation in training, and the corresponding objective function is defined as:

$$\mathcal{L} = \sum_{r_{i,t} \in R_i} -log(P(R_{i,1:t-1}|H_i, K_i)) \quad (2)$$

## 5 Experiments

### 5.1 Settings

**Model** Baselines consist of: **1) T5 and 2) mT5:** the monolingual English T5 (*t5-large, 738M*) (Raffel et al., 2020) and the multi-lingual mT5 (*mt5-base, 582M*) (Xue et al., 2021); **3) BART and 4) mBART:** the monolingual English BART (*bart-large, 406M*) (Lewis et al., 2020) and Chinese (*fnlp-bart-large, 407M*) (Shao et al., 2021) BART, as well as the multi-lingual mBART *(mbart-large, 611M)* (Liu et al., 2020); **5) Bert2Bert, 6) M-Bert2Bert, 7) CausalBert and 8) M-CausalBert:** models proposed by *XPersona* (Lin et al., 2020b), which use encoder-decoder models (5 and 6, *220-356M*) or autoregressive models (7 and 8, *356M*) for response generation. The prefix **M-** denotes it is a multi-lingual model, whereas a monolingual model. **Implementation:** We use the official codes for 5-8 models. Other models are implemented by PyTorch and Transformers. Similar to *XPersona*, we fine-tune each model with 5 epochs. We use the learning rate 1e-4 for T5/mT5-based models while 1e-5 for other models. The mini-batch size is 8, and the gradient accumulation step is 4. Experiments were run on Nvidia-A100 or RTX3090.

**Metrics** We adopt three representative metrics: **1) ES:** We first use SentenceBERT[3] to get the embedding of the ground-truth response and the generated response, and then use cosine function to compute the embedding similarity to estimate the semantic relevance; **2) F1:** we use the CharF1 to

---

[3]*paraphrase-multilingual-MiniLM-L12-v2*

compute the overlapping-based relevance and the precision of the generated response at the same time; **3) B4:** We use the 4-gram BLEU to estimate the quality of the generated response. Besides, we use **Mean-A** to estimate the overall performance among multiple languages. It first computes the local geometric mean score of three metrics in each language and then computes the weighted (the size of the test size is regarded as a weight) mean score over all local scores. **Mean-E** excludes the Indonesian. We do not use *perplexity* because of the notable vocabulary differences among models.

### 5.2 Main Results

As shown in Table 3, the multi-lingual knowledge-aware *MCK-T5*s can significantly outperform the monolingual knowledge-aware *MCK-T5*s, especially in the low-resource languages (*Ko*, *Id*, see *Native-ER* in Figure 2), demonstrating the effectiveness of using the multi-lingual commonsense knowledge. Meanwhile, due to insufficient native knowledge, *MCK-T5+moCsk* is beaten by the persona-aware *mT5+Pers*, indicating that monolingual commonsense knowledge can not provide information comparable to persona knowledge. Differently, *MCK-T5+Csk* uses multi-lingual commonsense knowledge and the overall performance is comparable to *mT5+Pers*. This result is incredible because the construction of such dialogues is totally grounded on persona knowledge rather than commonsense knowledge, and *MCK-T5+Csk* can be applied to more scenarios but *mT5+Pers* can only be used in the persona-aware scenario. Besides, the best *XPersona* baseline, *M-CausalBERT*, is worse than *mT5+Pers* and our *MCK-T5+Csk*. We think the reason is the backbone PLM of *M-CausalBERT* is worse than *mT5*.

For a more fair comparison in the persona scenario, *MCK-T5+Csk+Pers* is allowed to access the persona knowledge. In this case, our *MCK-T5* undoubtedly achieves the best performance in every dimension. This result demonstrates commonsense knowledge is essential in dialogue generation and would not conflict with other knowledge.

### 5.3 Native-Code Comparisons

Here, both the dialogue and the knowledge are in the same language. We test 3 configurations: 1) *Base* uses the *Native* sub-graph; 2) *Ext* additionally uses the *Transferred* sub-graph; and 3) *Tri* uses *Native+Transferred+Translated* sub-graphs. We also test the proposed two fact selection methods

| | *Know* | En | | | Cn | | | Fr | | | Jp | | | Ko | | | It | | | Id | | | Mean | |
|---|---|---|---|---|---|---|---|---|---|---|---|---|---|---|---|---|---|---|---|---|---|---|---|---|
| | Usage | ES | F1 | B4 | ES | F1 | B4 | ES | F1 | B4 | ES | F1 | B4 | ES | F1 | B4 | ES | F1 | B4 | ES | F1 | B4 | E | A |
| *BART* | *None* | 31.4 | 65.5 | 6.6 | 35.9 | 29.0 | 4.8 | - | - | - | - | - | - | - | - | - | - | - | - | - | - | - | - | - |
| *mBART* | *None* | 30.6 | 63.0 | 6.0 | 36.0 | 29.2 | 5.3 | 34.8 | 63.1 | 6.0 | 34.1 | 36.6 | 3.5 | 31.8 | 41.2 | 7.1 | 33.3 | 61.5 | 2.7 | 33.8 | 65.4 | 3.9 | 4.51 | 4.50 |
| *mBART* | *Pers* | 33.9 | 64.4 | 7.7 | 38.5 | 31.0 | 6.6 | 36.5 | 64.7 | 7.9 | 36.3 | 37.0 | 4.4 | 35.5 | 42.9 | 8.8 | 36.7 | 63.3 | 5.0 | 36.6 | 67.7 | 6.0 | 5.10 | 5.10 |
| *T5* | *None* | 31.5 | 65.4 | 6.7 | - | - | - | - | - | - | - | - | - | - | - | - | - | - | - | - | - | - | - | - |
| *mT5* | *None* | 30.9 | 65.8 | 6.7 | 35.7 | 29.3 | 5.3 | 31.9 | 63.6 | 5.9 | 34.6 | 38.3 | 3.6 | 29.7 | 42.4 | 6.1 | 33.2 | 64.5 | 3.8 | 30.3 | 64.6 | 4.1 | 4.62 | 4.56 |
| *mT5* | *Pers* | 33.3 | 66.8 | 8.3 | 38.5 | 31.3 | 6.9 | 35.9 | 65.6 | 8.2 | 36.5 | 38.2 | 4.9 | 35.3 | 44.0 | 9.8 | 37.2 | 65.7 | 6.5 | 34.7 | 68.1 | 5.9 | 5.29 | 5.27 |
| *Bert2Bert* | *Pers* | 29.8 | 63.3 | 5.4 | 33.3 | 26.9 | 3.1 | 24.0 | 61.6 | 2.4 | 32.5 | 35.0 | 1.5 | 23.7 | 38.4 | 3.2 | 31.6 | 62.0 | 2.2 | 23.7 | 65.1 | 1.5 | 3.80 | 3.63 |
| *M-Bert2Bert* | *Pers* | 32.9 | 62.9 | 4.2 | 36.8 | 23.1 | 1.2 | 35.3 | 63.3 | 2.7 | 34.7 | 29.9 | 0.4 | 33.0 | 38.7 | 2.9 | 35.7 | 60.9 | 1.2 | 36.0 | 65.4 | 1.2 | 3.54 | 3.45 |
| *CausalBert* | *Pers* | 32.2 | 62.7 | 7.7 | 35.0 | 29.0 | 4.5 | 34.1 | 62.8 | 5.6 | 34.1 | 36.4 | 3.0 | 31.8 | 41.1 | 6.5 | 36.2 | 60.6 | 4.2 | 34.0 | 65.4 | 4.6 | 4.69 | 4.69 |
| *M-CausalBert* | *Pers* | 32.9 | 63.3 | 7.7 | 36.8 | 30.0 | 5.0 | 35.3 | 63.7 | 5.8 | 34.7 | 36.5 | 3.1 | 33.0 | 42.1 | 6.7 | 35.7 | 60.7 | 4.1 | 36.0 | 66.2 | 4.9 | 4.78 | 4.80 |
| *MCK-T5* | *MoCsk* | 34.0 | 65.8 | 8.3 | 38.3 | 30.5 | 6.4 | 35.1 | 65.4 | 7.5 | 36.9 | 38.7 | 4.5 | 31.5 | 42.6 | 7.8 | 36.0 | 64.0 | 4.7 | 30.3 | 64.6 | 4.1 | 5.12 | 4.97 |
| *MCK-T5* | *Csk* | 34.6 | 65.7 | 8.6 | 38.7 | 31.1 | 6.1 | 36.2 | 65.7 | 8.1 | 37.8 | 38.5 | 4.7 | 34.7 | 43.8 | 9.3 | 36.2 | 64.9 | 5.3 | 35.1 | 67.0 | 5.4 | 5.27 | 5.23 |
| *MCK-T5* | *MoCsk+Pers* | 35.4 | 67.0 | 9.6 | 38.3 | 30.1 | 7.0 | 37.0 | 66.4 | 9.3 | 38.3 | 38.9 | 5.7 | 35.1 | 43.8 | 9.6 | 36.0 | 65.6 | 6.6 | 34.7 | 68.1 | 5.9 | 5.55 | 5.48 |
| *MCK-T5* | *Csk+Pers* | **38.2** | **67.5** | **10.2** | **41.5** | **32.8** | **7.7** | **39.0** | **66.7** | **9.6** | **39.5** | **39.6** | **5.8** | **38.6** | **45.6** | **11.2** | **39.3** | **66.0** | **8.0** | **39.2** | **68.7** | **7.6** | **5.87** | **5.87** |

Table 3: Model Comparisons. In *Know* column, *None* uses no knowledge, *Pers* uses the persona provided by *XPersona*, *Csk* uses 30 commonsense knowledge facts retrieved by the best configuration *'Soft Match, Tri, ER+RtT+TtR, Joint Training'* found in Table 5, *MoCsk* only uses the monolingual native commonsense knowledge.

| | En | | | Cn | | | Fr | | | Jp | | | Ko | | | It | | | Id | | | Mean | |
|---|---|---|---|---|---|---|---|---|---|---|---|---|---|---|---|---|---|---|---|---|---|---|---|
| | ES | F1 | B4 | ES | F1 | B4 | ES | F1 | B4 | ES | F1 | B4 | ES | F1 | B4 | ES | F1 | B4 | ES | F1 | B4 | E | A |
| *Vanilla (mT5)* | 30.9 | 65.8 | 6.7 | 35.7 | 29.3 | 5.3 | 31.9 | 63.6 | 5.9 | 34.6 | 38.3 | 3.6 | 29.7 | 42.4 | 6.1 | 33.2 | 64.5 | 3.8 | 30.3 | 64.6 | 4.1 | 4.62 | 4.56 |
| *Base+HM* | 33.6 | 66.0 | 8.0 | 37.0 | 29.9 | 6.1 | 34.1 | 65.3 | 7.4 | 35.5 | 38.3 | 4.5 | 31.0 | 42.7 | 7.9 | 34.2 | 64.9 | 4.6 | - | - | - | 5.04 | 4.90 |
| *Ext+HM* | 33.4 | 65.6 | 8.2 | 37.5 | 30.0 | 6.1 | 35.2 | 65.1 | 7.9 | 36.7 | 38.8 | 4.7 | 32.3 | 42.7 | 8.1 | 34.8 | 65.0 | 4.8 | - | - | - | 5.13 | 4.97 |
| *Tri+HM* | 33.6 | 65.9 | 8.3 | 37.5 | 30.4 | 6.1 | 34.6 | 65.2 | 7.6 | 36.7 | 38.7 | 4.6 | 33.2 | 43.5 | 8.8 | 35.4 | 64.9 | 5.1 | 33.2 | 66.5 | 5.2 | 5.16 | 5.10 |
| *Base+SF* | 34.0 | 65.8 | 8.3 | 38.3 | 30.5 | 6.4 | 35.1 | 65.4 | 7.5 | 36.9 | 38.7 | 4.5 | 31.5 | 42.6 | 7.8 | 36.0 | 64.0 | 4.7 | - | - | - | 5.12 | 4.97 |
| *Ext+SF* | 34.3 | 65.6 | 8.2 | 37.7 | 30.3 | 6.2 | 34.1 | 65.3 | 7.3 | 37.9 | 39.0 | 4.9 | 34.0 | 42.9 | 8.4 | 35.3 | 64.6 | 4.9 | - | - | - | 5.16 | 5.00 |
| *Tri+SF* | 34.3 | 66.0 | 8.1 | 38.0 | 30.6 | 6.3 | 34.7 | 65.3 | 7.7 | 37.3 | 38.9 | 4.8 | 33.4 | 43.6 | 8.7 | 35.5 | 65.2 | 5.1 | 33.0 | 66.4 | 4.9 | **5.17** | **5.10** |

Table 4: Native-code knowledge comparisons. Models use the *Singleton* training. Except for *Vanilla*, *HM* randomly selects 20 facts for each dialogue session while *SF* select the top 20 most relevant.

| | En | | | Cn | | | Fr | | | Jp | | | Ko | | | It | | | Id | | | Mean | |
|---|---|---|---|---|---|---|---|---|---|---|---|---|---|---|---|---|---|---|---|---|---|---|---|
| | ES | F1 | B4 | ES | F1 | B4 | ES | F1 | B4 | ES | F1 | B4 | ES | F1 | B4 | ES | F1 | B4 | ES | F1 | B4 | E | A |
| *Vanilla (mT5)* | 30.9 | 65.8 | 6.7 | 35.7 | 29.3 | 5.3 | 31.9 | 63.6 | 5.9 | 34.6 | 38.3 | 3.6 | 29.7 | 42.4 | 6.1 | 33.2 | 64.5 | 3.8 | 30.3 | 64.6 | 4.1 | 4.62 | 4.56 |
| *TtR* | 34.3 | 65.3 | 8.5 | 37.2 | 29.6 | 5.8 | 34.3 | 65.0 | 7.2 | 36.3 | 38.0 | 3.9 | 33.2 | 42.9 | 8.2 | 35.5 | 64.8 | 4.4 | 34.1 | 66.2 | 5.0 | 5.09 | 5.04 |
| *RtT* | 34.3 | 65.3 | 8.5 | 38.0 | 29.6 | 5.4 | 33.9 | 64.6 | 6.8 | 36.5 | 38.3 | 3.9 | 33.3 | 43.0 | 8.6 | 34.5 | 64.8 | 4.0 | 32.0 | 65.8 | 4.2 | 5.08 | 4.97 |
| *ER+TtR* | 34.3 | 66.0 | 8.1 | 37.9 | 30.1 | 5.7 | 34.4 | 65.1 | 7.5 | 36.8 | 38.6 | 4.6 | 33.6 | 43.1 | 8.6 | 34.7 | 65.0 | 4.8 | 33.3 | 65.6 | 5.0 | 5.12 | 5.06 |
| *ER+RtT* | 34.3 | 66.0 | 8.1 | 37.8 | 30.4 | 6.1 | 34.1 | 65.1 | 7.4 | 37.1 | 38.7 | 4.7 | 33.5 | 43.4 | 8.7 | 35.3 | 64.7 | 4.6 | 32.8 | 66.2 | 4.8 | 5.13 | 5.05 |
| *ER+RtT+TtR* | 34.3 | 66.0 | 8.1 | 38.2 | 30.4 | 6.2 | 35.2 | 64.6 | 7.6 | 37.2 | 38.6 | 4.8 | 35.0 | 43.3 | 8.8 | 35.6 | 64.1 | 4.8 | 32.4 | 65.8 | 4.6 | 5.17 | 5.10 |
| *ER+RtT+TtR+Joint* | 34.4 | 65.6 | 8.3 | 38.1 | 30.9 | 6.3 | 35.9 | 65.4 | 7.9 | 37.4 | 38.5 | 4.7 | 35.0 | 43.8 | 8.9 | 36.4 | 65.0 | 5.2 | 34.9 | 66.7 | 5.2 | **5.22** | **5.17** |

Table 5: Mixed-code knowledge comparisons. Except for *Vanilla*, models use the English and/or the native *Tri* sub-graphs, and use *SF* to select the 20 facts. Models use the *Singleton* training except for the last uses *Joint*.

*Hard Match (HM)* and *Soft Match (SF)*. Table 4 reports the results, and findings are discussed in the following paragraphs.

**The necessity of knowledge.** Compared to the non-knowledge *Vanilla*, all knowledge-aware models have achieved notable gains. It clearly indicates that 1) commonsense knowledge is necessary for open-domain dialogue response generation, and 2) our commonsense knowledge construction and alignment methods are at least not ineffective.

**The necessity of native-code multi-lingual knowledge.** Besides the *Native*, *Ext* and *Tri* use the knowledge transferred/translated from other languages. We can find *Ext* is better than *Base* and *Tri* is the best no matter if considering the Indonesian or not. It shows that using knowledge of other languages is an effective way to alleviate the knowledge shortage in the native language in both the high-resource and low-resource scenarios.

**The necessity of semantic-driven alignment.** *Soft-Match* can further consider the semantics during the alignment. The results have indicated this necessity because *SF* is always better than *HM*. Meanwhile, the performance gain of *SF* becomes weaker when using all three sub-graphs. This is because we only adopt the non-fine-tuned Sentence-BERT. We believe the more advanced fine-tuned ranker, such as cross/poly-encoders (Urbanek et al., 2019; Humeau et al., 2019; Wu et al., 2022a), can take a step further and leave this as future work.

| Chinese Case 1 | 你好,你有素食午餐食谱吗? Hello, do you have any vegan lunch recipes? |
|---|---|
| w/o. Csk | *Response:* 是的,我喜欢素食午餐。你呢? Yes, I like vegetarian lunch. What about you? |
| w/ Chinese Csk | (0) (好看 good looking, Causes, 喜欢 like); (1) (午餐 lunch, PartOf, 生活 life); 
 *Response:* 是的,我喜欢素食午餐。你呢? Yes, I like vegetarian lunch. What about you? |
| w/ TtR English Csk | (0) (animal, RelatedTo, horse); (1) (brown, IsA, color); (2) (brown, MannerOf, cook); 
 (3) (color, RelatedTo, set); .... (19) (vegan, IsA, vegetarian) 
 *Response:* 是的,我喜欢素食主义者。你呢? Yes, I like Vegetarianism. What about you? |
| w/ Multi-lingual Csk | (0) (晚饭 dinner, RelatedTo, 喜欢 like); (1) (最爱 favourite, DerivedFrom, 喜欢 like) .... 
 (18) (vegetarianism, RelatedTo, animal) (22)(vegetarian, RelatedTo, vegan) 
 *Response:* 不,我不是素食主义者,但我喜欢做饭。 No, I'm not a Vegetarianism, but I like cooking. |
| English Case 2 | Hello, do you have any vegan lunch recipes? |
| w/ English Csk | (0) (eat, RelatedTo, dinner); ... (8)(animal, Desires, eat); (9)(love animal, HasLastSubevent, eat); 
 *Response* No, I don't. I love animals. Do you? |
| w/ Multi-lingual Csk | (0) (enjoy, CausesDesire, eat); (1) (dinner, CapableOf, eat); ....(7) (cook, MotivatedByGoal, eat); 
 *Response* No, I don't. Do you like to cook? |

Table 6: Case Study. We can only list a few relevant facts because of the space. entities are the translated entities.

## 5.4 Mixed-Code Comparisons

We also consider using the knowledge in other languages without transferring or translating.

**English Only** We assume only English knowledge is available. For other languages, to use the cross-lingual English knowledge in a mixed-code way, *TtR* first translates dialogues to English and then recognizes the corresponding English entities; on the contrary, *RtT* first recognizes the native language entities and then translates them to English. As reported in Table 5, we can find that both *TtR* and *RtT* can significantly outperform the non-knowledge *Vanilla*, demonstrating 1) knowledge can be directly transferred from one knowledge to another knowledge in the open-domain dialogue response generation; 2) comparing such two models, *TtR* is a little better. We think the reason is the translation-first *TtR* can reduce the error distortion between the entity recognition process and the knowledge alignment process.

**English Plus** Then, we assume commonsense sub-graphs of both the native language knowledge and English are available. We can find that the corresponding *ER+TtR*, *ER+RtT*, and *ER+RtT+TtR* are better than the English-only *TtR* and *RtT*, showing more knowledge is better. Compared to *ER+RtT+TtR*, the last *ER+RtT+TtR+Joint* further uses the *Joint Training* strategy to train all instances of different languages in a uniform model. It can be seen that performance has increased significantly, indicating the knowledge of a specific language can be implicitly transferred to another language via joint training and a multi-lingual backbone language model.

## 5.5 Case study

Table 6 has shown two real cases sampled from our *MCK-Dialog* dataset. In the first Chinese case, four responses are generated with different knowledge configurations. The first **w/o. csk** does not use any knowledge; thus, the response is just a simple repetition. The second **w/. Chinese csk** only uses Chinese knowledge in the original ConceptNet. We find there is no enough relevant knowledge facts that can be used. Thus, the response is unchanged. The next **w/ TtR English csk** only uses the English knowledge in the ConceptNet via the *TtR* and knows what is *vegan*. However, the information is still not enough. The last **w/ Multi-lingual csk** uses the mixed-code multi-lingual knowledge; thus, the response is most relevant and informative. In the next case, we select an English case that is parallel to the previous Chinese case. Unlike the first case, no *vegan*-related fact has appeared because 1) there are too many general English facts, and our *Soft Match* is difficult to put *vegan*-related facts at the top; 2) we only use facts of the native language and English in this experiment, this makes English the least profitable of all languages. Nevertheless, the generated dialogues are still acceptable.

## 6 Future Work and Conclusion

### 6.1 Future Work

This work focuses on proposing the task and constructing the dataset/baselines. Based on the foundation, there can be some new research topics:

*1) Knowledge Selection:* As mentioned, there are 800/89K+ *matched/related* facts in each dialogue session. Obviously, a model is difficult to handle such a large number of facts, and a response would not involve so many facts. Thus, how to select

proper facts are important.

*2) Knowledge Conflicts:* Using knowledge from multiple sources and languages simultaneously would inevitably lead to knowledge conflicts, such as translation errors, repetitions across languages, and many others. Eliminating these conflicts is necessary for further improving efficiency.

## 6.2   Conclusion

Prior commonsense knowledge-aware dialogue response generation works have achieved promising results. However, they only study high-resource languages such as English and Chinese. Differently, this paper focuses on both the high-resource and low-resource languages. Hence, we first propose a new *Multi-Lingual Commonsense Knowledge-Aware Response Generation (MCKRG)* task, which tries to use commonsense knowledge in other languages to enhance dialogue generation in the current language. Then, we construct a dataset *MCK-Dialog* of seven languages and a baseline model *MCK-T5*. Extensive experimental results demonstrate the great potential of using multi-lingual commonsense knowledge in high-resource and low-resource languages. To our best knowledge, this work is the first to explore *MCKRG*.

## Acknowledgement

This work is supported in part by the National Natural Science Foundation of China under Grant 62162067 and 62101480, Research and Application of Object detection based on Artificial Intelligence, in part by the Yunnan Province expert workstations under Grant 202205AF150145.

## Limitation

This work studies the multi-lingual commonsense knowledge-aware dialogue response generation. Besides the future work mentioned in Section 6.1, there are still some limitations in the work.

**Benchmark Dataset**   *MCK-Dialog* only targets to be a benchmark dataset because the construction mainly relied on automatic methods, which may impact the quality besides the English dialogues. This can be attributed to the backbone conversational corpus *XPersona* was built in a similarly automatic way. Nevertheless, we still believe that building benchmark datasets in an automated manner is still necessary for multi-lingual studies, as the cost of manual construction is too high, especially when the number of languages studied increases.

**Human Evaluation**   This work does not conduct a human evaluation. The reason is two-fold: 1) This work involves seven different languages, posing great challenges in employing volunteers, designing the human evaluation process and quality control; 2) We have conducted an early human evaluation for English and Chinese dialogues, we find this evaluation may not have enough persuasiveness. Non-English conversations were translated from English, and their texts are quite different from human-written texts, even though they have been revised by humans (Lin et al., 2020b).

## Ethical Consideration

This paper studies a widely studied topic, namely, open-domain dialogue response generation, and uses open-released resources. We do not introduce any new ethical statement/consideration in this work. Consequently, we think this work has no additional ethical considerations compared to previous work from the technical perspective.

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
