# OpenReview forum: "Exploring the Effectiveness of Multi-Lingual Commonsense Knowledge-Aware Open-Domain Dialogue Response Generation"
_EMNLP/2023/Conference — EMNLP 2023 Findings_

### Official Review · Reviewer_z5FY · 2023-08-04

**Soundness:** 4

**Excitement:**

3: Ambivalent: It has merits (e.g., it reports state-of-the-art results, the idea is nice), but there are key weaknesses (e.g., it describes incremental work), and it can significantly benefit from another round of revision. However, I won't object to accepting it if my co-reviewers champion it.

**Paper Topic And Main Contributions:**

This paper constructs multi-lingual MCKRG dataset MCK-Dialog and proposes a new model MCK-T5 to flexibly integrate multiple knowledge sources into dialogue generation. Specifically, the new method use commonsense knowledge in other languages to enhance the current dialogue generation. At last, the method is tested on the XPersona dataset and proven to achieve noticeable improvements.

**Reasons To Accept:**

1. The paper is well-organized, contains enough information in a limited number of pages, and is easy to understand.
2. The paper evaluates different variants of the method and most of them achieve better results than baselines. The experiments in the paper are extensive and convincing. The chosen evaluation metrics are presentative, and ablation experiments show the efficacy of the new method.
3. The paper is one of the first to explore the potential of introducing multi-lingual commonsense graph to dialogue generation.
4. According to the experiment results, the new method can effectively boost model’s performance in low-resource languages such as Indonesian.

**Reasons To Reject:**

1. I did not find any major reasons to reject this paper.

**Reproducibility:**

4: Could mostly reproduce the results, but there may be some variation because of sample variance or minor variations in their interpretation of the protocol or method.

**Reviewer Confidence:**

4: Quite sure. I tried to check the important points carefully. It's unlikely, though conceivable, that I missed something that should affect my ratings.

**Typos Grammar Style And Presentation Improvements:**

On line 504, “ER+RtT+TTR” and “ER+RtT+TTR+Joint” should be “ER+RtT+TtR” and “ER+RtT+TtR+Joint”.

---

### Official Review · Reviewer_wvbY · 2023-08-04

**Soundness:** 4

**Excitement:**

3: Ambivalent: It has merits (e.g., it reports state-of-the-art results, the idea is nice), but there are key weaknesses (e.g., it describes incremental work), and it can significantly benefit from another round of revision. However, I won't object to accepting it if my co-reviewers champion it.

**Paper Topic And Main Contributions:**

The paper proposes the new task of multilingual commonsense knowledge-based dialogue response generation. The authors argue that commonsense knowledge is effective in improving dialogue response generation by reducing non-informative and factually incorrect generations.

Using XPersona and ConceptNet as their base dialogue and commonsense knowledge datasets, the authors note that commonsense knowledge is relatively scarce in languages other than English. They build the MCK-Dialog dataset by augmenting the knowledge triples in each language by 1) transferring triples from other languages using ConceptNet's cross-lingual Synonym relation and 2) translating triples from other languages using NMT. The triples are then aligned to utterances and aggregated by dialogue session.

The authors then train dialogue response generation models, based on multilingual T5, using knowledge from both the dialogue language and the other six languages in MCK-Dialog. The experimental results show that models using multilingual commonsense knowledge perform on par with baseline models using persona knowledge (despite the base dialogue dataset focusing on personas), and that combining multilingual commonsense knowledge with personas gives the best performance. The authors perform an extensive analysis; other major findings include the fact that triplets transferred and translated from other languages improve performance, even for English and the other high-resource languages, and training one model on all languages performs better than training separate models for different dialogue languages.

**Questions For The Authors:**

What source language is used for the translated knowledge facts? Is it a mix of all other available languages, or is it just English like in the alignment step? Do you perform any deduplication of translated facts from different source languages?

Can you explain why the number of Native-RtT facts is so high for the non-English languages? It makes sense that it is higher than Native-ER, since the non-English languages have smaller graphs in the original ConceptNet, but why is it so much higher than Native-TtR? Is this a translation noise issue? Section 5.4 mentions that TtR performs better than RtT, even though there are more RtT facts. Is this a knowledge selection issue?

I may have missed this information in the text, but I cannot find what knowledge selector is used in the experiments. Section 3.4 lines 330-331 mention that turn-level knowledge selection is needed because the MCK-Dialog dataset contains facts grounded at the dialogue level.

I am also curious as to why the dataset is grounded at the dialogue level? From Section 3.3, it seems to me that the knowledge alignment approach could produce utterance-level knowledge alignments simply by not aggregating the session entities. This would have the benefit of 1) not requiring a knowledge selection step when experimenting on the response generation step and 2) allowing for the evaluation of multilingual knowledge selectors.

**Reasons To Accept:**

The proposed task is well-motivated; the dataset construction process is sensible, and the authors compare several different options for augmenting the fact triplets and aligning them to the dialogues. The experimental results show that the proposed multilingual approach outperforms monolingual approaches, even in high resource languages.

**Reasons To Reject:**

The proposed approaches are straightforward and make use of existing methods (eg. translate-then-retrieve, retrieve-then-translate). However, I think this is fine because the authors perform an extensive analysis comparing these settings in their experimental results.

A minor issue is that the experimental results seem to be confounded by an unknown knowledge selection step (see questions below).

Update: My concerns were addressed in the author rebuttal, and I have raised my score accordingly.

**Reproducibility:**

4: Could mostly reproduce the results, but there may be some variation because of sample variance or minor variations in their interpretation of the protocol or method.

**Reviewer Confidence:**

3: Pretty sure, but there's a chance I missed something. Although I have a good feel for this area in general, I did not carefully check the paper's details, e.g., the math, experimental design, or novelty.

**Typos Grammar Style And Presentation Improvements:**

Typos:
Section 1, line 046: CocneptNet -> ConceptNet
Figure 1: Brid -> Bird
Section 2, line 166: generation -> translation
Section 5.1, line 393: ER -> ES
Section 5.1, line 408: perpleixty -> perplexity
Section 5.2, lime 433: CausualBERT -> CausalBERT
Section 5.5, line 520: no -> not

Both "Cn" and "Zh" are used as abbreviations for Chinese; using just one consistently would be more clear for the reader.

Overall there are many awkward phrasings; the paper would benefit from another round of editing.

---

### Official Review · Reviewer_cLnM · 2023-08-04

**Soundness:** 3

**Excitement:**

3: Ambivalent: It has merits (e.g., it reports state-of-the-art results, the idea is nice), but there are key weaknesses (e.g., it describes incremental work), and it can significantly benefit from another round of revision. However, I won't object to accepting it if my co-reviewers champion it.

**Paper Topic And Main Contributions:**

This paper is the first to utilize multi-lingual commonsense knowledge for enhancing dialogue generation. It constructs a dataset in 7 languages through translation, and proposes a novel baseline model MCK-T5 based on mT5. Evaluation of this model on the newly collected dataset reveals that the incorporation of knowledge from high-resource languages can significantly enhance the dialogue systems for low-resource languages.

**Reasons To Accept:**

- This paper presents a multilingual dialogue dataset, known as MCK-Dialog, which aims to enhance dialogue systems of low-resource languages using commonsense knowledge from high-resource knowledge.
- The paper introduces MCK-T5 as the baseline model ,and conducts a comprehensive analysis of the effects of each design element on the overall performance.

**Reasons To Reject:**

- There is no comparison with translate-test baseline, which translates dialogue to English, generates a response using an English system, and then translates back to the original language. This baseline is straightforward (only use off-the-shelf English system and NMT system) and has been proven to be one of the most competitive approach in various multilingual NLP tasks.

- Table 5 illustrates that combining knowledge from both English and the native language only slightly outperforms the English-only variants. In terms of TtR and RtT, the average improvement in overall performance is a marginal 0.03% and 0.05%, respectively. Therefore, based on these results, the argument for using multiple languages instead of solely relying on English, a high-resource language, is less compelling.

- No human evaluation. Many studies have demonstrated a discrepancy between automatic evaluation results and human preferences on dialogue generation tasks.

- (Minor) As mentioned in section 3.3.2, the proposed system rely on knowledge from two languages, i.e., the native language and English. Consequently, it might be more accurate to describe the task as a "cross-lingual" knowledge-aware dialogue, rather than "multilingual".

**Reproducibility:**

4: Could mostly reproduce the results, but there may be some variation because of sample variance or minor variations in their interpretation of the protocol or method.

**Reviewer Confidence:**

5: Positive that my evaluation is correct. I read the paper very carefully and I am very familiar with related work.

---

### Meta-Review · Area_Chair_kfZY · 2023-09-12

**Recommendation:** 4

**Metareview:**

This paper proposes to use multilingual commonsense knowledge to augment dialogue generation. It creates a dataset spanning seven languages through translation and introduces a baseline model, MCK-T5, built upon the foundation of mT5. An evaluation of this model on the curated dataset demonstrates a notable enhancement in dialogue systems for low-resource languages when high-resource language knowledge is integrated.

While the reviewers did express several concerns about the experiments, the majority of these concerns have been adequately addressed in the rebuttal.

---

### Meta-Review · Senior_Area_Chairs · 2023-10-02

**Recommendation:** 4

**Metareview:**

This paper proposes to use multilingual commonsense knowledge to augment dialogue generation. It creates a dataset spanning seven languages through translation and introduces a baseline model, MCK-T5, built upon the foundation of mT5. An evaluation of this model on the curated dataset demonstrates a notable enhancement in dialogue systems for low-resource languages when high-resource language knowledge is integrated.

While the reviewers did express several concerns about the experiments, the majority of these concerns have been adequately addressed in the rebuttal.

---

### Decision · Program_Chairs · 2023-10-07

**Decision:**

Accept-Findings

**Comment:**

This paper proposes to use multilingual commonsense knowledge to augment dialogue generation. It creates a dataset spanning seven languages through translation and introduces a baseline model, MCK-T5, built upon the foundation of mT5. An evaluation of this model on the curated dataset demonstrates a notable enhancement in dialogue systems for low-resource languages when high-resource language knowledge is integrated.

While the reviewers did express several concerns about the experiments, the majority of these concerns have been adequately addressed in the rebuttal.|This paper proposes to use multilingual commonsense knowledge to augment dialogue generation. It creates a dataset spanning seven languages through translation and introduces a baseline model, MCK-T5, built upon the foundation of mT5. An evaluation of this model on the curated dataset demonstrates a notable enhancement in dialogue systems for low-resource languages when high-resource language knowledge is integrated.

While the reviewers did express several concerns about the experiments, the majority of these concerns have been adequately addressed in the rebuttal.